# The Impact of COVID-19 Pandemic on Weight Loss, Eating Behaviour and Quality of Life after Roux-en-Y Gastric Bypass

**DOI:** 10.3390/medicina59091597

**Published:** 2023-09-04

**Authors:** Karolina Bauraitė, Rita Gudaitytė, Almantas Maleckas

**Affiliations:** 1Department of Gastroenterology, Medical Academy, Lithuanian University of Health Science, 44307 Kaunas, Lithuania; 2Department of Surgery, Medical Academy, Lithuanian University of Health Science, 44307 Kaunas, Lithuania; rita.gudaityte2@lsmuni.lt (R.G.); almantas.maleckas@lsmu.lt (A.M.); 3Department of Gastrosurgical Research and Education, Sahlgrenska Academy, University of Gothenburg, 405 30 Gothenburg, Sweden

**Keywords:** bariatric surgery, Roux-en-Y gastric bypass, COVID-19, weight loss, eating behaviour, quality of life

## Abstract

*Background and Objectives*: The global pandemic of coronavirus disease (COVID-19), declared on 11 March 2020, had an extensive impact on bariatric patients. The aim of this study was to evaluate short-term weight loss outcomes, changes in eating behaviour, and health-related quality of life (HRQoL) among patients who had Roux-en-Y gastric bypass (RYGB) before and during the COVID-19 pandemic. *Materials and Methods*: This cohort study included 72 patients (Group S) who underwent RYGB surgery in the Surgery Department of the Lithuanian University of Health Sciences during the COVID-19 pandemic in the years 2020–2022. Data for the control group (Group C) of 87 patients (operated on in 2010–2012) were collected from a prospective study. The data referred to the period before and a year after the RYGB. The information about patients’ weight changes, hunger, satiety, fullness sensations, appetite, diet, and eating patterns was queried. Eating behaviour and HRQoL evaluation were conducted by the Three-Factor Eating Questionnaire (TFEQ-R18) and the medical outcomes study Short-Form-36 (SF-36), respectively. *Results*: One year after the surgery, % excess body mass index loss (%EBMIL) was 77.88 (26.33) in Group S, 76.21 (19.98) in Group C, *p* = 0.663. Patients in Group S tended more to choose snacks between main meals: 79.2% versus 28.7%, *p* < 0.0001. Cognitive restraint significantly increased in Group S from 45.93 (13.37) up to 54.48 (13.76), *p* = 0.001; additionally, significantly worse overall health status was found in Group S compared to Group C, 53.27 (24.61) versus 70.11 (31.63), *p* < 0.0001. Mental HRQoL (50.76 versus 60.52 score, *p* < 0.0001) and social functioning (44.79 versus 57.90, *p* < 0.0001) were worse in Group S. *Conclusions*: In this study, the COVID-19 pandemic had no impact on short-term weight loss after RYGB. However, one year after, RYGB patients tended to snack more, and mental HRQoL and social functioning were worse in the study group.

## 1. Introduction

The global pandemic of coronavirus disease 2019 (COVID-19), declared on 11 March 2020 by The World Health Organization (WHO), had an extensive impact on the population [1]. Mandatory lockdowns caused sudden lifestyle changes that triggered behaviour related to weight gain [2,3]. Eating out of stress or boredom, more temptations to eat, and higher consumption of low nutritional value foods were negative impacts that led to adverse results for not only general population but also for patients with obesity who were trying to reach their goals [4].

During the last four decades, the rate of obesity disease has raised three times [5]. Unfortunately, if the current tendency is maintained, the numbers will continue to increase. Obesity is the result of a positive energy balance: energy intake is higher than expenditure. Factors such as genetics, hormonal issues, medications, etc., might influence weight changes [6]. Moreover, obesity might cascade to other serious diseases, such as hypertension, diabetes, some cancer, mental health issues, impaired immune system, etc. The pandemic years showed that COVID-19-positive patients with obesity were more likely to be hospitalized, and the risk is three times higher than for normal body mass index (BMI) patients [7]. Thus, diagnosing and treating obesity might be the key to preventing or reversing the comorbidities, extending life expectancy, and improving the quality of life [8].

Bariatric surgery is known as the most effective treatment for morbid obesity and related chronic diseases, which leads to improved patient quality of life [9]. It was also proven that patients, after bariatric surgery, had reduced morbidity and mortality from COVID-19 infection [10]. Due to the superb weight loss results, one of the most popular methods is Roux-en-Y gastric bypass (RYGB) [11]. The first two years after RYBG is the period when a patient is expected to reach the results by losing excess weight and forming habits that lead to sustainable weight loss.

Nevertheless, the coronavirus pandemic had a negative impact on postoperative bariatric surgery patients due to disruption in dietary and physical activity routines, barriers to obtaining effective follow-up care, and increased stress [12]. There are contradictory data about short-term weight loss among patients who underwent bariatric surgery before and during the COVID-19 pandemic. Two studies [13,14] have found that patients operated on during the COVID-19 pandemic lost significantly less weight, while four studies did not find any difference [15,16,17,18]. However, during the COVID-19 pandemic, more maladaptive eating behaviour, leading to loss of control overeating and poor dietary quality, was observed among the patients after bariatric surgery [19]. Emotional eating and losing daily lifestyle routines were only a few triggering factors related to social isolation experienced during the COVID-19 pandemic years. Eating psychopathology was related to worsening of mental-health-related quality of life [20]. Thus, further studies on changes in weight loss, eating behaviour, and health-related quality of life during the COVID-19 lockdowns may improve our understanding of the possible outcomes of bariatric surgery in this vulnerable group of patients. The aim of the current study was to evaluate short-term weight loss outcomes, changes in eating behaviour, and health-related quality of life (HRQoL) among patients who had RYGB before and during the COVID-19 pandemic. The hypothesis that the COVID-19 pandemic negatively impacted the bariatric population was raised.

## 2. Materials and Methods

### 2.1. Study Design and Participants

The data used in this study refer to the period before the surgery and the year after the bariatric surgery. Study participants were divided into two independent groups. The study group (Group S) included patients who had RYGB surgeries during the COVID-19 pandemic in the years 2020–2022. Overall, 115 patients underwent RYGB in 2020–2022, and 43 were excluded from the study. Twenty-eight patients skipped the appointments, fourteen patients did not respond to the phone call or email, and one patient did not agree to fill out questionnaires. There were 72 (62.61%) subjects in the study group who agreed to participate in the conducted research. The control group (Group C) included 87 patients who had RYGB surgery in the years 2010–2012. The RYGB technique was the same as during the COVID-19 period. Data for this group were collected from a prospective study performed in 2011–2013 [21].

According to standard follow-up protocol, patients came, or were invited by phone or e-mail, to the outpatient clinic for an annual check-up. During the one-year follow-up, the information about patients’ weight changes, hunger and satiety sensations, feelings of fullness (the feeling while the stomach is full, but would like to continue eating), appetites, diets, and eating patterns were queried. They indicated if they have hunger between meals and satiety after meals using the Visual Analogue Scale (VAS), which was rated on a 10-grade scale, where 0 was no hunger or satiety and 10 was extreme hunger or satiety. The standardized dietary questionnaire after gastric bypass surgery was used, and the patients were questioned if they experienced the feeling of fullness after their meals (after each meal, a few times a day, once a day, once a week, or did not have this feeling), how big was the portion size they could eat in comparison to preoperative portions (the same, three-quarters preoperative portion, half of preoperative, one-quarter of preoperative, very small) and if they were having snacks between main meals [8]. Patients described their changes in appetite (did not change; decreased; disappeared; disappeared, but some of the days it was coming back), the eating pleasure (daily, two–three times a week, once a week, two–three times a month, a few times per year, did not feel). The Three-Factor Eating Questionnaire (TFEQ-R18) and the medical outcomes study Short-Form-36 (SF-36) questionnaires helped to collect standardized information about eating behaviour and HRQoL. Before data collection, information about confidentiality was provided. After patients agreed to participate in this study, questionnaires were provided. Percent excess BMI loss (%EBMIL) was calculated using the formula: [pre-operative BMI kg/m^2^ − current BMI kg/m^2^]/[pre-operative BMI kg/m^2^ − 25 kg/m^2^] × 100 [22].

The TFEQ-R18 questionnaire measures three separate aspects of eating behaviour: uncontrolled eating (a loss of control of eating because of subjective feelings of hunger that lead to eating more than usual), cognitive restraint (a constant restriction of food intake to maintain body weight or to induce the weight loss), and emotional eating (an inability to resist food due to emotional stimulus). The TFEQ-R18 tool includes 18 items, each question has a response scale, and the answers are scored from 1 to 4 [23]. Scores have been summated into separate parts for uncontrolled eating (UE), which includes nine items, cognitive restraint (CR), and emotional eating (EE), which include six and three items, respectively. The raw scale scores were transformed to a 0–100 scale = [((raw score − lowest possible raw score)/possible raw score range) × 100] [21]. Higher scores on the TFEQ-R18 represent more uncontrolled eating, cognitive restraint, and emotional eating.

The SF-36 survey was used to measure HRQoL. This item is divided into eight subscales: physical functioning (PF), role limitations due to physical health (RP), role limitations due to emotional problems (RE), vitality (V), social functioning (SF), bodily pain (BP), general health (GH), mental health (MH), and one self-reported item on health change. The SF-36 represents two major demotions of health: physical and mental. The PF, RP, and BP and MH, RE, and SF subscales measure the physical and mental dimensions, respectively. The V and GH subscales measure both. Thus, two orthogonal summary scores are extracted: the physical component summary (PCS) and the mental component summary (MCS) [24,25]. The raw scale scores were transformed to a 0–100 scale, and a result of 100 is kept as the best HRQoL. Transformed scale = [((actual raw score − lowest possible raw score)/possible raw score range) × 100] [25]. The higher scores indicate better health status.

### 2.2. Surgical Procedure

RYGB was performed by the same surgeon using an identical surgical approach (Figure 1). A gastric pouch of 15 to 30 mL was constructed using linear staplers. The biliopancreatic limb of 120–150 cm was measured using the hand-over-hand technique under medium stretch along the mesenteric border. An antecolic antegastric end-to-side gastrojejunostomy, 30 mm wide, was performed with the combined method (linear stapler and hand-sewn). The length of the alimentary bowel loop ranged from 100 to 120 cm. Side-to-side entero-enteroanastomosis was performed in the same way as gastrojejunostomy. The afferent loop of the small bowel was divided between entero-enteroanastomosis and gastrojejunostomy to create Roux-en-Y gastric bypass [8].

### 2.3. Ethics

This study was approved by the regional biomedical research ethics committee of Kaunas (P1-BE-2-88/2021).

### 2.4. Statistical Analysis

Microsoft Excel 365 was used for data collection and TFEQ-R18, SF-36 scoring. Data were analysed with the IBM SPSS Statistics version 23 program (IBM Corp., Armonk, NY, USA). The results of the assessment forms are presented as means (standard deviations). The equality of the variances between the two populations was assessed with the *f*-test and the unpaired *t*-test (to compare the differences in the means between the groups). Chi-square (χ^2^) was used for binary data. The statistical difference was determined as *p* < 0.05.

## 3. Results

There were no statistically significant differences between baseline measures (age, gender, BMI before the RYGB surgery) in the two study groups (Table 1). The average BMI before the RYGB surgery in Group S was 46.07 (8.42) and there was no statistically significant difference in comparison with Group C—44.95 (6.57), *p* = 0.359. The %EBMIL one-year after surgery was 77.88 (26.33) and 76.21 (19.98), respectively, *p* = 0.663. Percentage of total weight loss (%TWL) in the study group reached 33.03 (9.37), and in the control group it reached 36.32 (17.59), *p* = 0.135. Significantly more patients in Group S had cardiovascular diseases, type 2 diabetes mellitus, musculoskeletal disorders, gastrointestinal, and respiratory diseases (Table 1).

The TFEQ-R18 and SF-36 questionnaires have been used to evaluate eating behaviour and HRQoL at baseline and one year after the RYGB in both groups. There were no significant differences between the groups in TFEQ-R18 scores one year after the bariatric surgery. Significantly higher CR score was found at baseline in the Group C as compared to Group S: 53.53 (14.06) versus 45.93 (13.37), *p* = 0.001 (Table 2). Nevertheless, one year after the RYGB, CR significantly increased in Group S from 45.93 (13.37) up to 54.48 (13.76), *p* = 0.001. There was a significant decrease in UE and EE in both groups (*p* < 0.0001) one year after the RYGB surgery (Figure 2 and Figure 3).

The overall health status score estimated by SF-36 significantly improved in both groups: The Group S score increased from 19.17 (23.64) to 53.17 (24.61), *p* < 0.0001, and, in Group C, the score increased from 29.22 (20.59) to 70.11 (31.63), *p* < 0.0001. A year after the RYGB surgery, patients in Group S had significantly worse overall health status as compared to Group C: 53.27 (24.61) versus 70.11 (31.63), *p* < 0.0001. Before RYGB surgery, SF, GH, and PCS scores of Group S were significantly worse than Group C (Table 2). One year after RYGB, significantly worse scores in Group S were found in the RE, V, SF, MH, BP, PCS, and MCS domains. When the delta scores of all domains were compared between the groups, significantly higher increases in RE, MH, and MCS scores were observed in Group C and in GH score in Group S (Table 2). The SF and V scores decreased significantly in Group S and significantly increased in Group C one year after RYGB (Figure 4 and Figure 5). Both groups presented significantly improved PCS score results: Group S from 50.17 (16.59) to 69.26 (10.95), *p* < 0.0001 and Group C from 55.96 (12.87) to 74.04 (11.58), *p* < 0.0001. There was no significant change of MCS score in Group S: from 49.73 (8.15) to 52.10 (8.66), *p* = 0.11. In Group C, a significant MCS increase was observed: from 50.65 (9.50) to 61.69 (11.08), *p* < 0.0001.

One year after the RYGB, among Group S patients, the average satiety score after meals was 6.47 (2.87) as compared to 7.79 (1.61), *p* = 0.002 among the patients in Group C. The lack of the feeling of satiety after meals was reported by 20.8% in Group S and 13.1% in Group C (*p =* 0.200). There was no difference in the hunger scores between the meals among the groups: 4.97 (2.92) and 5.31 (1.82) in Group S and Group C, respectively, *p* = 0.43. In Group S, 51.4% of the patients one year after surgery declared a decreased appetite, as opposed to 79.8% in the Group C, *p* = 0.0002. No feeling of eating pleasure was reported by 12.5% in Group S and 29.8% in Group C (*p* = 0.009). The portion sizes were similar between the groups; 70.8% in Group S and 81% in Group C reported eating one-quarter or less of the preoperative portion size (*p =* 0.137). In addition, patients in Group S tended to choose more snacks between main meals: 79.2% versus 28.7%, *p* < 0.0001.

## 4. Discussion

The present study investigated the impact of the COVID-19 pandemic lockdowns on weight loss, eating behaviour, and HRQoL after RYGB surgery. The results indicate that bariatric surgery performed during the pandemic years had similar weight loss as compared to the pre-pandemic period. CR increased, and UE and EE scores significantly decreased one year after the surgery in both groups. Satiety was less expressed, and more people felt pleasure while eating in the study group as compared to the control group. Moreover, study group patients tended to snack more. This study demonstrates that the role limitation due to emotional problems, vitality, mental health, and body pain were worse for patients who had surgeries during the COVID-19 pandemic period. Also, as could be expected, social functioning was worse in the study group.

The COVID-19 pandemic period had no influence on short-term weight loss results after RYGB in our patient population, and this finding is supported by the observations from the other published studies [15,16,17]. The %EBMIL one year after RYGB in our study was 77.9 and 76.2 among the patients operated on during the pandemic and before the pandemic, respectively. Similarly, Pereira X et al. [18] found %EBMIL of 71.8 and 70.1 one year after RYGB in COVID-19 affected and COVID-19 unaffected groups, respectively. A possible explanation could be that RYGB itself has a significant impact on eating behaviour. We observed that, one year after RYGB, patients in the study group had a significant increase in CR eating score and a significant decrease in UE and EE scores. Similar results were seen among patients undergoing surgery in the pre-pandemic period. Thus, the COVID-19 pandemic had no influence on eating behaviour after RYGB. An opposite trend was observed in the general population during the pandemic—dimensions of UE and EE increased without change in CR [26]. CR behaviour is associated with reduced caloric intake to control body weight or improve body image [27]. Even during the COVID-19 pandemic, young adults who had higher CR scores achieved weight loss [28]. Increased stress during the COVID-19 pandemic, negative body image, and unhealthy eating habits are factors associated with EE and UE [27]. The three domains of eating behaviour interact. Increased CR after RYGB in our study population may lead to healthier eating habits, weight loss, and improved body image, and result in reduced EE and UE.

During the COVID-19 lockdown, food became a coping mechanism for dealing with emotions. Athanasiadis et al. presented a survey study that described the negative effect of the lockdown period on patients after bariatric surgery. It was found that 48.2% of patients lost eating control, and a tendency towards increased snacking (62.6% of the bariatric patients) was observed [29]. In our study, snacking was reported by 79.2% in the pandemic group as compared to 28.7% in the pre-pandemic group, even though UE score decreased significantly one year after surgery in both groups. Increased snacking between meals could be defined as a coping mechanism, as mastication could reduce stress and induce changes in the central nervous system, especially in the hippocampus and hypothalamus [30].

Based on a survey conducted by Hu A et al., loss of control eating was negatively associated with satiety responsiveness [19]. Our findings presented that patients’ satiety feelings and decreased appetites were less manifested, and pleasure while eating was felt more often in the study group. Consequently, these changes could be the trigger for choosing more snacks. A positive correlation between emotional overeating and enjoyment of food was established [19]. However, Youssef A et al., in their nested-qualitative study, found that patients during the peak period of the pandemic experienced eating as momentary pleasure [20]. EE and UE scores among our patients after RYGB were reduced significantly, suggesting that snacking was not associated with loss of control eating, and pleasure while eating was one of the mechanisms to cope with stress. However, in the future, such eating behaviour may lead to increased weight regain. A study by Andreu A. et al. showed that EE and the time after bariatric surgery were statistically significant risk factors for predicting weight gain [31]. Conceicao E. et al. found that weight regain 3 years after the surgery was higher during the pandemic period as compared to the results before the pandemic [32]. In our study, one year weight loss after RYGB was not affected, but patients operated on during the pandemic ate snacks more often, had reduced satiety, fewer experienced decreased appetite, and more enjoyment with food was observed. Future studies should determine what influence such changes in eating behaviour may have on long-term weight loss and weight regain.

Lockdowns and increased levels of distress due to COVID-19 have a significant impact on HRQoL. A Swiss general population study found that physical HRQoL significantly increased while mental HRQoL significantly decreased during the COVID-19 pandemic [33]. By contrast, in a Spanish population, both components, physical and mental, were significantly lower during the pandemic [34]. To our knowledge, this study was the first to investigate HRQoL using the SF-36 questionnaire among the patients undergoing bariatric surgery during a pandemic. The results of our study show that patients who had RYGB during the pandemic period had a significant decrease in social functioning and vitality on year after surgery. Significant improvement was observed in physical functioning, limitations due to physical health, general health, bodily pain, and PCS. Mental components, such as limitations due to emotional problems, mental health, and MCS, improved insignificantly. The pre-pandemic RYGB cohort showed significant improvement in both physical and mental HRQoL. However, we have observed a decrease in GH score from 60.3 to 55.4 in the latter cohort. As pre-pandemic patients were younger and had fewer concomitant diseases, their expectations regarding postoperative general health could have been higher, and these did not always match. The stressful environment during the pandemic had a significant impact on mental HRQoL. Mental HRQoL did not decrease among our patients one year after RYGB during the pandemic, as was shown in the Swiss and Spanish general population; however, the patients did not experience a significant positive impact of bariatric surgery on mental HRQoL, as is usually expected. What impact the lifting of COVID-19 restrictions may have on the mental HRQoL of bariatric patients operated on during the pandemic still has to be investigated.

Our study has several limitations. First, during the lockdowns, in our institution, non-essential elective surgical procedures, including bariatric surgery, were cancelled and, later, gradually reinstalled, prioritizing bariatric patients with higher BMI and co-morbidities. Thus, our study population had more concomitant diseases before RYGB, and this could have had an impact on the SF-36 scores, such as for general health results. To avoid such selection bias, we estimated the delta scores of all of the SF-36 subscales and compared them between the groups. Second, due to restrictions on outpatient consultations during the lockdowns period, some of the data were collected through phone calls and e-mail. Moreover, some of the anthropometric variables was self-reported. It was considered a common limitation due to the experienced epidemiological situation [27]. Third, the outpatient consultations during lockdowns were cancelled and, later, the majority of follow-up dietary consultations for bariatric patients were provided online. We did not investigate if this had an impact on the adherence to follow-up and dietary recommendations. Moreover, there are no data on whether counselling on eating behaviour after bariatric surgery is equally efficient for online and face-to-face consultations. However, there is some evidence that intuitive eating interventions on eating behaviours delivered online or face-to-face have a similar effect [35]. Finally, we did not investigate what impact COVID-19 infection had on HRQoL. However, no patients in our cohort, when interviewed, had acute or long COVID-19 symptoms.

## 5. Conclusions

In this study, the COVID-19 pandemic had no impact on short-term weight loss after RYGB. However, one year after RYGB, in the group operated on during the pandemic, more patients were snacking, and mental HRQoL and social functioning were worse. Further studies should be conducted to evaluate what impact the pandemic years may have on long-term eating behaviour, mental HRQoL, and weight loss results.

## Figures and Tables

**Figure 1 medicina-59-01597-f001:**
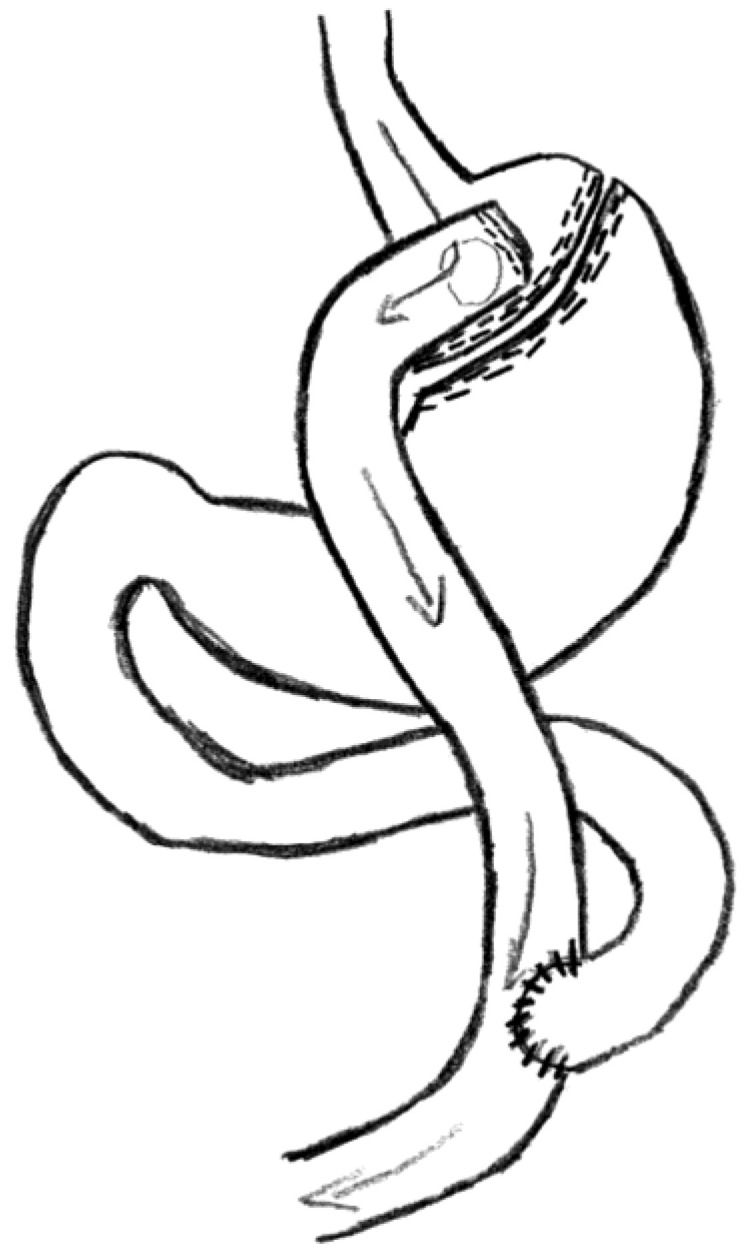
Roux-en-Y gastric bypass (RYGB). The figure was made by article co-author Rita Gudaitytė.

**Figure 2 medicina-59-01597-f002:**
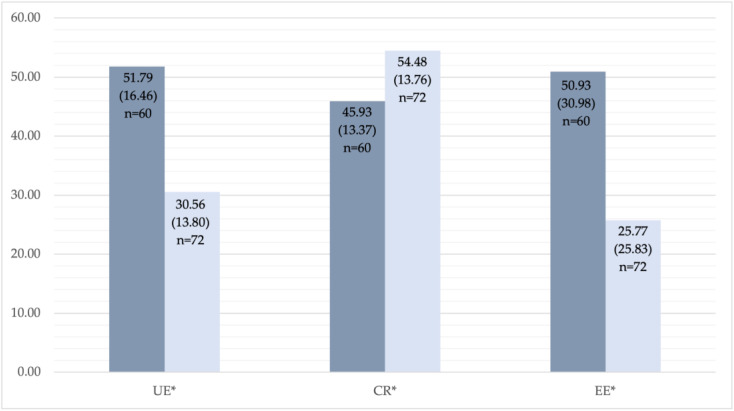
TFEQ (the Three-Factor Eating Questionnaire)—R18 results, study group: before and one year after RYGB. * Statistically significant changes: uncontrolled eating (UE), *p* < 0.0001; cognitive restraint (CR), *p* = 0.001; emotional eating (EE), *p* < 0.0001. The dark column represents results before the RYGB, the light column represents results one year after the RYGB.

**Figure 3 medicina-59-01597-f003:**
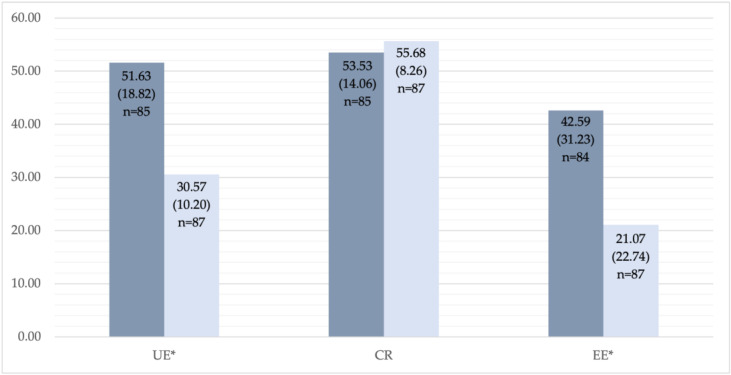
TFEQ (the Three-Factor Eating Questionnaire)—R18 results, control group: before and one year after RYGB. * Statistically significant changes: uncontrolled eating (UE), *p* < 0.0001; cognitive restraint (CR), *p* = 0.272; emotional eating (EE), *p* < 0.0001. The dark column represents results before the RYGB, the light column represents results one year after the RYGB.

**Figure 4 medicina-59-01597-f004:**
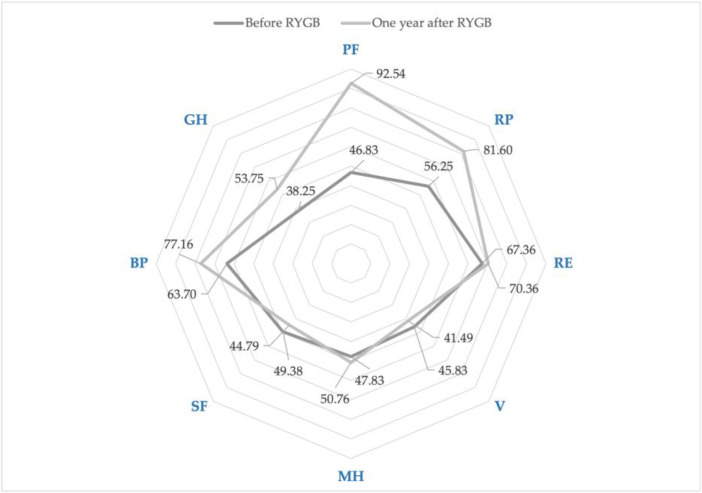
SF-36 changes: before and one year after the RYGB, study group. Physical functioning (PF) *p* < 0.0001; role limitations due to physical health (RP), *p* < 0.0001; role limitations due to emotional problems (RE), *p* = 0.568; vitality (V), *p* = 0.037; mental health (MH), *p* = 0.132; social functioning (SF), *p* = 0.026; bodily pain (BP), *p* = 0.012; general health (GH), *p* < 0.0001.

**Figure 5 medicina-59-01597-f005:**
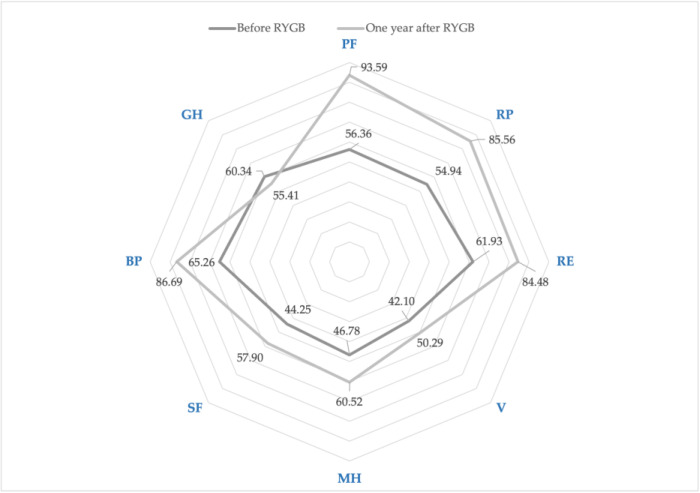
SF-36 changes: before and one year after the RYGB, control group. Physical functioning (PF) *p* < 0.0001; role limitations due to physical health (RP), *p* < 0.0001; role limitations due to emotional problems (RE), *p* < 0.0001; vitality (V), *p* < 0.0001; mental health (MH), *p* < 0.0001; social functioning (SF), *p* < 0.0001; bodily pain (BP), *p* < 0.0001; general health (GH), *p* = 0.025.

**Table 1 medicina-59-01597-t001:** Characteristics of the study groups.

	Group S * (*n* = 72)	Group C (*n* = 87)	*p* Value
Age before the surgery, yr., mean (SD)	44.31 (11.95)	41.91 (10.98)	0.189
Sex F/M	51/21	62/25	0.107
BMI before the surgery, mean (SD)	46.07 (8.42)	44.95 (6.57)	0.359
%EBMIL one yr. post-surgery, %, (SD)	77.88 (26.33)	76.21 (19.98)	0.663
Cardiovascular diseases ** (%)	32.20 ^1^	12.99	0.005
Hypertension ** (%)	69.49 ^1^	65.52	0.618
Diabetes ** (%)	28.81 ^1^	10.34	0.004
Musculoskeletal disorders ** (%)	37.29 ^1^	17.24	0.007
Gastrointestinal diseases ** (%)	27.12 ^1^	1.15	<0.0001
Respiratory disease ** (%)	20.34 ^1^	6.90	0.016
Depression ** (%)	25.00 ^2^	35.63	0.173
Work style ** (%):			
Not working	6.78 ^1^	0.00	0.014
Sedentary	50.85 ^1^	40.23	0.207
Variable	19.00 ^1^	37.93	0.015
Not sedentary	10.17 ^1^	18.39	0.175
Intense physical work	0.00 ^1^	3.45	0.121

* Patients, who had Roux-en-Y gastric bypass surgery during the COVID-19 pandemic. ** Before Roux-en-Y gastric bypass surgery. ^1^
*n* = 59, ^2^
*n* = 60. BMI = body mass index, kg/m^2^, %EBMIL = % excess BMI loss (excess BMI > 25 kg/m^2^), SD = standard deviation.

**Table 2 medicina-59-01597-t002:** TFEQ-R18 and SF-36 scoring results: before and one year after RYGB surgery.

	Before RYGB			One Year after RYGB			Change from Baseline to Follow-Up		
Variables	Group S *	Group C	*p* Value	Group S	Group C	*p* Value	Delta	Delta Group C	*p* Value
	(*n* = 60)	(*n* = 87)		(*n* = 72)	(*n* = 87)		Group S		
TFEQ (mean, (SD))									
Uncontrolled eating	51.8 (16.5)	51.6 (18.8) ^1^	0.959	30.6 (13.8)	30.6 (10.2)	0.996	−20.9 (20.0)	−21.0 (20.3)	0.978
Cognitive restraint	45.9 (13.4)	53.5 (14.1) ^1^	0.001	54.5 (13.8)	55.7 (8.3)	0.515	7.4 (20.6)	2.2 (15.7)	0.099
Emotional eating	50.9 (31.0)	42.6 (31.2) ^2^	0.115	25.8 (25.8)	21.1 (22.7)	0.225	−26.7 (31.7)	−21.2 (35.0)	0.335
SF-36 (mean, (SD))									
Physical functioning	46.8 (32.3)	56.4 (26.2) ^3^	0.064	92.5 (12.0) ^4^	93.6 (14.5) ^1^	0.627	46.1 (31.9)	37.8 (27.4)	0.102
Role limitations due to physical health	56.3 (30.5)	54.9 (23.4) ^3^	0.781	81.6 (24.7) ^4^	85.6 (23.1)	0.300	27.5 (34.2)	29.9 (27.9)	0.660
Role limitations due to emotional problems	67.4 (32.3)	61.9 (23.7) ^3^	0.273	70.4 (27.5) ^5^	84.5 (24.3)	0.001	3.9 (42.0)	21.9 (31.5)	0.007
Vitality	45.8 (9.6)	42.1 (14.0)	0.057	41.5 (13.9)	50.3 (12.5)	<0.0001	−4.6 (15.8)	8.2 (20.0)	<0.0001
Mental health	47.8 (10.8)	46.8 (14.6)	0.616	50.8 (11.3)	60.5 (14.2)	<0.0001	3.3 (16.6)	13.7 (21.1)	0.001
Social functioning	49.4 (10.9)	44.3 (15.9)	0.022	44.8 (12.2)	57.9 (17.8)	<0.0001	−5.0 (16.8)	13.7 (25.8)	<0.0001
Body pain	63.7 (32.9)	65.3 (27.1)	0.762	77.2 (26.4)	86.7 (20.8)	0.014	14.6 (36.0)	21.5 (27.4) ^6^	0.218
General health	38.3 (14.7)	60.3 (17.0)	<0.0001	53.8 (14.4)	55.4 (11.0) ^6^	0.424	16.9 (18.9)	−5.1 (19.8) ^6^	<0.0001
PCS	50.2 (16.6)	56.0 (12.9)	0.025	69.3 (11.0)	74.0 (11.6)	0.009	20.0 (16.9)	18.1 (14.1)	0.456
MCS	49.7 (8.2)	50.7 (9.5)	0.542	52.1 (8.7)	61.7 (11.1)	<0.0001	2.6 (11.7)	11.1 (15.3)	0.0002

* Patients, who had Roux-en-Y gastric bypass surgery during the COVID-19 pandemic. ^1^
*n* = 85, ^2^
*n* = 84, ^3^
*n* = 81, ^4^
*n* = 71, ^5^
*n* = 70, ^6^
*n* = 86. TFEQ—the Three-Factor Eating Questionnaire. SF-36—the 36-item short form survey. SD—standard deviation. PCS—physical component summary, MCS—mental component summary.

## Data Availability

Not applicable.

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
