# Peer review of "The Impact of COVID-19 Pandemic on Weight Loss, Eating Behaviour and Quality of Life after Roux-en-Y Gastric Bypass"

_medicina, 2023, doi:10.3390/medicina59091597_

Round 1

Reviewer 1 Report

Dear authors,

I have read and evaluated your research in detail. First of all, I would like to thank you for your contribution to the literature. I think it is appropriate for your research to be published in the journal "Medicina" after the corrections I have mentioned below.

Specific Comments

Introduction

This chapter is fluid and well written. State your hypothesis clearly only after the purpose statement.

method

This section looks very messy. Please divide this section into sections such as design, participants, procedures. Also, give all the tests and scales you used with titles. Finally, please provide information about the surgical model you have applied and a photograph, if any.

Discussion

This section does not need any revision.

Author Response

Response to Reviewer 1 Comments

Dear Reviewer,

Thank you for giving us the opportunity to submit a revised manuscript titled “The Impact of COVID-19 Pandemic on Weight Loss, Eating Behaviour and Quality of Life After Roux-En-Y Gastric Bypass” to the special issue “Obesity and Bariatric Surgery: Updates and Challenges" by “Medicina” journal consideration. We appreciate the time and effort that have dedicated to providing your valuable feedback on our manuscript. The changes have been made and highlighted within the manuscript.

Here is a point-by-point response to the comments:

Point 1:

“Introduction

This chapter is fluid and well written. State your hypothesis clearly only after the purpose statement.”

Response 1: Thank you, we stated and added the hypothesis at the end of the introduction.

Point 2:

“Method

This section looks very messy. Please divide this section into sections such as design, participants, procedures. Also, give all the tests and scales you used with titles. Finally, please provide information about the surgical model you have applied and a photograph, if any.”

Response 2:

  • It is a great remark, the part of the Materials and Methods was modified and divided into sections: Study Design and Participants, Surgical procedure, Ethics and Statistical Analysis.
  • The information about used tests and scales was updated.
  • The information about surgical procedures was put in a separate section (2. Surgical procedure). The manuscript has been supplemented with the additional figure.

Point 3:

“Discussion

This section does not need any revision.”

Response 3: Thank you for such a positive comment and for pointing this out.

We look forward to hearing from you regarding the submission and are more than pleased to respond to any further comments and questions you may have.

Sincerely,

Karolina BauraitÄ—, M.D.

Department of Gastroenterology,

Medical Academy, Lithuanian University of Health Sciences

Reviewer 2 Report

Very interesting article. Apparently, the operated on patients suffered consequences related to COVID restrictions, like those suffering from other diseases. You presented some of these restrictions in the last paragraph of the "discussion". I congratulate you.

Author Response

Response to Reviewer 2 Comments

Dear Reviewer,

Thank you for giving us the opportunity to submit a revised manuscript titled “The Impact of COVID-19 Pandemic on Weight Loss, Eating Behaviour and Quality of Life After Roux-En-Y Gastric Bypass” to the special issue “Obesity and Bariatric Surgery: Updates and Challenges" by “Medicina” journal consideration. We appreciate the time and effort that have dedicated to providing your valuable feedback on our manuscript.

Here is a response to the comment:

Point 1: “Very interesting article. Apparently, the operated on patients suffered consequences related to COVID restrictions, like those suffering from other diseases. You presented some of these restrictions in the last paragraph of the "discussion". I congratulate you.”

Response 1: Thank you for such a positive response and for pointing this out.

We look forward to hearing from you regarding the submission and are more than pleased to respond to any further comments and questions you may have.

Sincerely,

Karolina BauraitÄ—, M.D.

Department of Gastroenterology,

Medical Academy, Lithuanian University of Health Sciences

Reviewer 3 Report

Authors investigated the influence of the pandemic of Covid-19 on the bariatric surgery effect . The aim had a novelty and the conclusions were seemed to be feasible, however there was a question on the interpretation of the results as below. 

・Authors described in the conclusion that ', one year after RYGB in the group operated during the pandemic more 331 patients were snacking, mental HRQoL and social functioning were worse',  however in Table 2 'Change from baseline to follow-up' of general health were 16.9 in  delta Group S and -5.1 in delta Group C.  The result seemsed to show worsening of general health in Group C.  Authors were recommended to speculate about this discrepancy more in detail in the discussion part. 

Author Response

Response to Reviewer 3 Comments

Dear Reviewer,

Thank you for giving us the opportunity to submit a revised manuscript titled “The Impact of COVID-19 Pandemic on Weight Loss, Eating Behaviour and Quality of Life After Roux-En-Y Gastric Bypass” to the special issue “Obesity and Bariatric Surgery: Updates and Challenges" by “Medicina” journal consideration. We appreciate the time and effort that have dedicated to providing your valuable feedback on our manuscript. The changes have been made and highlighted within the manuscript.

Here is a point-by-point response to the comments:

Point 1: “Authors investigated the influence of the pandemic of Covid-19 on the bariatric surgery effect. The aim had a novelty and the conclusions were seemed to be feasible, however there was a question on the interpretation of the results as below. 

  • Authors described in the conclusion that ', one year after RYGB in the group operated during the pandemic more 331 patients were snacking, mental HRQoL and social functioning were worse',  however in Table 2 'Change from baseline to follow-up' of general health were 16.9 in  delta Group S and -5.1 in delta Group C.  The result seemsed to show worsening of general health in Group C.  Authors were recommended to speculate about this discrepancy more in detail in the discussion part. “

Response 1: Thank you for such an insightful comment. Patients in Group S had more concomitant diseases and General Health measured by SF-36 before bariatric surgery was significantly lower as compared to Group C. This is in a line with our previous study where we found that baseline General Health score in type 2 diabetes patients in Lithuanian population was 37.8. However, one year after surgery there was no significant difference in General Health score between our groups. We agree that General Health score in Group C have decreased from 60 to 55, but it could be to the fact that this group was younger with less concomitant diseases and their expectations regarding postoperative general health have been much higher and not always matched.

Changes were made and we specified it in the Discussion part, 341-344 lines: “However, we have observed decrease in GH score from 60,3 to 55,4 in the latter cohort. As pre-pandemic patients were younger with less concomitant diseases their expectations regarding postoperative general health could have been higher and not always matched.” and in the limitations paragraph, 354-355 lines: “Thus, our study population had more concomitant diseases before RYGB and this could have had an impact on SF-36 scores, such as general health results”.

We look forward to hearing from you regarding the submission and are more than pleased to respond to any further comments and questions you may have.

Sincerely,

Karolina BauraitÄ—, M.D.

Department of Gastroenterology,

Medical Academy, Lithuanian University of Health Sciences